# Identification of Cuproptosis Clusters and Integrative Analyses in Parkinson’s Disease

**DOI:** 10.3390/brainsci13071015

**Published:** 2023-06-30

**Authors:** Moxuan Zhang, Wenjia Meng, Chong Liu, Huizhi Wang, Renpeng Li, Qiao Wang, Yuan Gao, Siyu Zhou, Tingting Du, Tianshuo Yuan, Lin Shi, Chunlei Han, Fangang Meng

**Affiliations:** 1Beijing Neurosurgical Institute, Capital Medical University, Beijing 100070, China; mxzhang91@163.com (M.Z.); liuch2012@163.com (C.L.);; 2Department of Neurosurgery, Beijing Tiantan Hospital, Capital Medical University, Beijing 100070, China; 3Beijing Key Laboratory of Neurostimulation, Beijing 100070, China; 4Clinical School, Tianjin Medical University, Tianjin 300270, China; mengwenjia1127@163.com; 5Chinese Institute for Brain Research, Beijing 102206, China

**Keywords:** Parkinson’s disease, cuproptosis, molecular subtypes, immune infiltration, LASSO analysis, prediction model

## Abstract

Parkinson’s disease (PD) is the second most common neurodegenerative disease; it mainly occurs in the elderly population. Cuproptosis is a newly discovered form of regulated cell death involved in the progression of various diseases. Combining multiple GEO datasets, we analyzed the expression profile and immunity of cuproptosis-related genes (CRGs) in PD. Dysregulated CRGs and differential immune responses were identified between PD and non-PD substantia nigra. Two CRG clusters were defined in PD. Immune analysis suggested that CRG cluster 1 was characterized by a high immune response. The enrichment analysis showed that CRG cluster 1 was significantly enriched in immune activation pathways, such as the Notch pathway and the JAK-STAT pathway. KIAA0319, AGTR1, and SLC18A2 were selected as core genes based on the LASSO analysis. We built a nomogram that can predict the occurrence of PD based on the core genes. Further analysis found that the core genes were significantly correlated with tyrosine hydroxylase activity. This study systematically evaluated the relationship between cuproptosis and PD and established a predictive model for assessing the risk of cuproptosis subtypes and the outcome of PD patients. This study provides a new understanding of PD-related molecular mechanisms and provides new insights into the treatment of PD.

## 1. Introduction

Parkinson’s disease (PD), which follows Alzheimer’s disease as the second most frequent neurodegenerative disorder, is characterized by the degeneration of dopaminergic neurons in the substantia nigra and striatum [1,2]. Approximately 7 million people worldwide are affected by PD, with the majority being elderly individuals [3]. Gender and age are identified as independent risk factors for PD [4]. It is estimated to affect approximately 1% of adults over the age of 60, and the prevalence rate over the age of 85 reaches 5%, causing a series of familial, medical, and social problems [5]. Although PD is largely considered to be a sporadic disorder, several gene mutations, including α-synuclein (SNCA), have been identified as causative genes for familial PD [6]. The common clinical manifestations of PD are divided into motor symptoms and non-motor symptoms, in which motor symptoms include static tremors, bradykinesia, myotonia, and postural balance disturbance, and in which non-motor symptoms include sleep disturbance, loss of smell, constipation, anxiety, depression, and cognitive decline [7]. In the early stages, PD symptoms tend to resemble those of various other conditions, which often complicates the clinical evaluation of the disease [8]. Due to its complex pathogenesis, the current main treatment for PD (levodopa therapy and DBS) still cannot completely control the symptoms and their progression [9]. Despite extensive efforts, clinical biomarkers for PD remain elusive due to its high heterogeneity.

Previous studies have shown that the development of PD is closely related to mitochondrial dysfunction, neuroinflammation, oxidative stress, and apoptosis [10]. Neuroinflammation is a typical pathological characteristic of PD. Neuroinflammation mediated by microglia, astrocytes, and peripheral immune cells exerts neurotoxic effects by exacerbating neuronal damage [11]. Postmortem brain analyses of PD patients have shown increased nucleic acid oxidation and lipid peroxidation in the substantia nigra and striatum, suggesting that excessive levels of reactive oxygen species and free radicals lead to neuronal damage and death [12]. DAergic neurons in the substantia nigra are vulnerable to mitochondrial DNA (mtDNA) damage due to oxidative stress and neuroinflammation, ultimately leading to mitochondrial dysfunction [13]. However, current treatments, as well as novel preventive and therapeutic measures, work similarly and indiscriminately for PD patients. Therefore, there is an urgent need to discover new diagnostic and therapeutic targets to improve the quality of life of PD patients.

Currently, molecular biomarkers have promising applications in the diagnosis and treatment of PD. Recently, Tsvetkov et al. revealed that copper accumulation disrupts mitochondrial metabolic enzymes, triggering a novel regulated cell death (RCD) mechanism called “cuproptosis” that differs from apoptosis, ferroptosis, and pyroptosis [14]. This study demonstrates that cuproptosis occurs when copper binds directly to lipidated proteins in the tricarboxylic acid (TCA) cycle, leading to acute proteotoxic stress, mitochondrial metabolic dysfunction, and, ultimately, cell death. Copper is an indispensable trace element involved in a variety of biological processes, and it plays a vital role in maintaining cellular enzyme activity [15,16]. Altered copper levels in the body may lead to oxidative stress and cytotoxicity and contribute to disease initiation and progression [17,18]. Previous studies have shown that copper is abnormally distributed in aging brain tissue and associated with neurodegenerative diseases [19,20]. Disrupted copper homeostasis can result from genetic mutations, aging, or environmental factors, and it can contribute to a range of pathological changes, including cancer and neurodegeneration [21]. An epidemiological study has suggested that chronic copper exposure in the workplace increases the risk of Parkinson’s disease [22]. Other studies have shown that copper excess can lead to neuronal cell death and α-synuclein aggregation [23]. In addition, many studies have shown that mitochondrial dysfunction and oxidative stress may play a key role in the progression of PD [24,25,26]. Nonetheless, the possible regulatory mechanisms of cuproptosis in PD are not yet fully understood. Therefore, the role of cuproptosis-related genes (CRGs) in the development of PD deserves further attention.

In this study, we conducted a comprehensive analysis of the expression disparities and immune features of CRGs between normal substantia nigra and PD substantia nigra. Based on the expression profiles of 13 CRGs, we performed the consensus clustering analysis of 81 PD patients and further evaluated the differences in immune infiltration and functional enrichment between the two clusters. Subsequently, the WGCNA algorithm was employed to identify DEGs related to cuproptosis and PD, and the enriched biological functions and pathways were subsequently determined based on these DEGs. We then selected three core genes through a variety of machine learning algorithms. Finally, we further validated the stability of the core genes in the test cohort in an attempt to provide new solutions for the treatment of PD.

## 2. Materials and Methods

### 2.1. Data Acquisition and Preprocessing

Expression profiles of PD (GSE7621, GSE20141, GSE49036, GSE20186, GSE20295, GSE8397, GSE26927, and GSE133101) were downloaded from the Gene Expression Omnibus (GEO; http://www.ncbi.nlm.nih.gov/geo/ (accessed on 13 November 2022)) database [27]. The GSE7621, GSE20141, GSE49036, GSE20186, GSE20295, and GSE8397 were normalized using the “normalizeBetweenArrays” function in the “limma” package. Subsequently, the “combat” function in the “sva” package was used to remove batch effects and to serve as the training set [28]. Overall, the training set totaled 81 PD substantia nigra samples and 65 control samples. For external validation, we selected two datasets as our testing set, including GSE26927 and GSE133101. We performed the same preprocessing on the testing set. The information of these datasets is listed in Table 1.

### 2.2. Assessment of Immune Cell Infiltration

The relative abundance of 22 immune cells in each sample was calculated using the “CIBERSORT” algorithm based on the LM22 signature matrix (https://cibersort.stanford.edu/ (accessed on 11 September 2022)) [29]. Only samples with deconvolution *p*-values < 0.05 were considered to accurately measure immune cell composition. The R package “ESTIMATE” can calculate immune scores using gene expression profiles [30]. The immune score for PD patients was evaluated between CRG clusters.

### 2.3. Unsupervised Clustering of Cuproptosis-Related Genes (CRGs)

A total of 13 CRGs were obtained from previous studies [14]. Utilizing the expression differences of CRGs, we conducted a consensus clustering analysis with the “ConsensusClusterPlus” R package, categorizing 81 PD samples into distinct groups [31]. The optimal number of clusters was determined by considering CDF curves, consensus matrices, and consensus cluster scores.

### 2.4. Gene Set Variation Analysis (GSVA) and Gene Set Enrichment Analysis (GSEA)

GSVA and GSEA enrichment analyses were performed to elucidate the differences in biological functionbetween different CRG clusters [32,33]. The annotation package was obtained from the MSigDB website (https://www.gsea-msigdb.org/ (accessed on 30 November 2022)). The results were visualized using the “heatmap” package in R. GSEA enrichment analysis was performed by GSEA software downloaded from the MSigDB website to explore the involved signaling pathways between CRG clusters.

### 2.5. Identification of DEGs between Different CRG Clusters and Functional Enrichment Analysis

The DEGs between different CRG clusters were identified using the “limma” package [34]. The cut-off values for screening DEGs were based on |log2FC| ≥ 1 and FDR < 0.05. Gene Ontology (GO) and Kyoto Encyclopedia of Genes and Genomes (KEGG) enrichment analyses were performed using the “clusterProfiler” package in R to explore the biological functions of CRG cluster-related DEGs [35].

### 2.6. Weighted Gene Co-Expression Network Analysis (WGCNA)

WGCNA analysis was conducted to identify co-expression modules using the “WGCNA” package [36]. WGCNA analysis was used to investigate the relationship between different modules and clinical characteristics. The independence and average connectivity of different modules were assessed with various power values (ranging from 1 to 20). To maintain the high reliability of the results, the minimum number of genes was set to 30. The “moduleEigengenes” function was employed to calculate the module eigengene for each model to identify key modules. Module Membership (MM) indicated the relationship between modules and disease status, while Gene Significance (GS) represented the correlation between genes and clinical phenotypes. Candidate hub genes within the module were selected using the criteria of |GS| > 0.2 and |MM| > 0.8.

### 2.7. Establishment of A Predictive Model Based on Machine Learning Methods

In order to narrow down the list of candidate genes and eliminate confounding factors, a least absolute shrinkage and selection operator (LASSO) analysis was conducted [37]. This analysis aimed to identify key genes that do not exhibit a relationship with each other, thus mitigating the risk of overfitting. The candidate genes identified through LASSO analysis in the training cohort were utilized to build a logistic regression model. This model aimed to investigate the correlation between disease occurrence and these genes. Furthermore, the model was assessed using ROC curve analysis to evaluate its sensitivity and specificity in both the training and testing cohorts.

### 2.8. Construction of a Nomogram

A nomogram model for evaluating the occurrence of PD was established using the “lrm” function of the “rms” package in R [38]. For each predictor, there is an associated score, and the cumulative score is employed to forecast the incidence of the disease. We used calibration curves to assess the predictive power of nomogram models.

### 2.9. Validation of the Model

The stability of the candidate genes were validated by ROC using two test sets (GSE26927 and GSE133101). The reliability of the model was judged by plotting the ROC and calculating the AUC area. A boxplot was used to analyze the expression of model genes in the test set. *p* < 0.05 was considered statistically significant.

### 2.10. Statistical Analysis

Each experiment was conducted a minimum of three times, and the results are expressed as the mean ± standard deviation (SD). Comparisons between two independent groups were made using the Student’s *t*-test. To evaluate the statistical significance among three or more groups, one-way ANOVA was employed. Statistical analysis of the data was performed using R software (version 4.2.2), Prism 8, and SPSS (version 27.0). A *p*-value of less than 0.05 was deemed to indicate statistical significance.

## 3. Results

### 3.1. Identification of Cuproptosis-Related Genes and Activation of Immune Responses in PD Patients

The research flowchart is shown in Figure 1. Our analysis included six GEO datasets containing 81 PD substantia nigra tissues and 65 control substantia nigra tissues. Based on previous studies, 13 genes (ATP7B, ATP7A, SLC31A1, FDX1, LIAS, LIPT1, DLD, DLAT, PDHA1, PDHB, MTF1, GLS, and CDKN2A) were confirmed to be associated with cuproptosis [11]. To confirm the role of the CRGs in PD, we determined the expression levels of these 13 CRGs between PD substantia nigra and normal substantia nigra using the GEO database. A total of eight CRGs were identified as differentially expressed genes in PD substantia nigra. Among them, the expression levels of MTF1 were higher, whereas ATP7A, FDX1, LIAS, DLD, DLAT, PDHB, and GLS gene expression levels were lower in PD substantia nigra tissues than in the control samples (Figure 2A–C). We conducted a correlation analysis, which revealed that DLD and DLAT exhibited a strong synergistic effect with a coefficient of 0.79. Simultaneously, GLS showed obvious antagonism with MTF1 (coefficient = 0.40) (Figure 2D). The gene association diagram further illustrated the close connections among these differentially expressed CRGs (Figure 2E). To elucidate whether there were variations in the immune system between the PD and the controls, the immune cell infiltration of substantia nigra was constructed through ESTIMATE algorithms (Figure 2F). The results revealed that PD substantia nigra presented higher infiltration levels of naive CD4+ T cells, resting memory CD4+ T cells, resting NK cells, Monocytes, M2 macrophages, and Neutrophils. Simultaneously, CD8+ T cells, follicular helper T cells, and active NK cells decreased in the substantia nigra of PD (Figure 2G). These findings imply that alterations within the immune system might be among the contributing factors to the development of PD. Meanwhile, the correlation analysis indicates that CRGs are correlated with immune infiltration (Figure 2H). These results suggest that CRGs may be associated with PD progression and immune infiltration.

### 3.2. Identification of Cuproptosis-Related Clusters in PD

To investigate the cuproptosis-related expression patterns in PD, we executed consensus clustering analysis focused on the eight CRGs utilizing the Consensus Cluster Plus package within R. When k = 2, we obtained the best cluster stability, and we finally divided 81 PD patients into two CRG clusters, including cluster 1 (*n* = 40) and cluster 2 (*n* = 41) (Figure 3A,B). The PCA plot displayed distinct distributions between the two CRG clusters (Figure 3C).

### 3.3. Correlations of Cuproptosis-Related Clusters with Immune Checkpoint Genes and Immune Microenvironment

To explore the molecular features between the two CRG clusters, we first assessed the differential expression of the eight CRGs between cluster 1 and cluster 2. Through the heatmap, we found that there were significant differences in the expression of CRGs between the two clusters (Figure 3D). Boxplot analysis showed that ATP7A, FDX1, LIAS, DLD, DLAT, PDHB, and GLS were highly expressed in cluster 2, while the expression of MTF1 was enhanced in cluster 1 (Figure 3E). To examine the role of CRGs in immune infiltration associated with PD, we compared the immune scores and immune infiltration levels between the two CRG clusters utilizing the ESTIMATE. The immune score of patients in cluster 1 was higher than that of cluster 2 (Figure 3F). The immune infiltration analysis revealed a change in the immune microenvironment between the two CRG clusters (Figure 3G). Then, we explored the correlation of clusters with immune checkpoint genes and immune score. Our analysis revealed that CRG cluster 1 exhibited a higher expression of immune checkpoint genes (Figure 3H).

### 3.4. Functional Annotation of CRG Clusters

To further explore the biological function between the two CRG clusters, we carried out GSVA, GO, and KEGG analyses. The GSVA of GO terms showed that CRG cluster 1 was significantly enriched in growth-related processes (growth factor binding, collagen metabolic process, and collagen containing extracellular matrix). Simultaneously, CRG cluster 2 was significantly enriched in synapse-related processes, including inhibitory synapse, neurotransmitter gated ion channel clustering, and proteasome binding (Figure 4A). The GSVA of KEGG terms revealed that CRG cluster 1 was significantly enriched in the Notch pathway, ECM receptor interaction, JAK-STAT pathway, and cytokine receptor interaction. CRG cluster 2 was significantly enriched in cell cycle, RNA degradation, ubiquitin-mediated proteolysis, pyruvate metabolism, and citrate cycle TCA cycle (Figure 4B). We then performed a GSEA analysis, which was consistent with GSVA. The GSEA results showed that CRG cluster 1 was enriched in cytokine receptor interaction, focal adhesion, and Notch signaling pathway (Figure 4C). CRG cluster 2 was significantly enriched in Parkinson’s disease, proteasome, and ubiquitin-mediated proteolysis, which is consistent with the GSVA (Figure 4D). Then, we explored the metabolic characteristics between the two clusters. The results showed that cluster 1 had a significant decline in energy metabolism (such as citric acid cycle, inositol phosphate metabolism, and oxidative phosphorylation) and biosynthesis (fatty acid elongation and Cholesterol Biosynthesis) (Figure 4E).

### 3.5. Identification of Cuproptosis Gene Subtypes in PD

To further explore the underlying biology of the CRG clusters, we performed differential analysis and identified transcriptome differences between the CRG clusters. A total of 77 DEGs were identified between the two CRG clusters using package “limma” in R. We performed consensus clustering of 77 DEGs obtained from the differential analysis, which divided PD patients into two gene clusters (Figure 5A,B). The PCA plot showed an obviously different distribution between the two gene clusters (Figure 5C). The heatmap delineated the DEGs between the two gene clusters, and the DEGs that were positively correlated with the gene cluster were classified as gene type A, while the rest of the DEGs were named gene type B (Figure 5D). Through the heatmap, we found that the gene cluster was consistent with the CRG cluster. We explored the correlation between gene clusters and CRGs, and the results were generally consistent with the results of CRG clusters (Figure 5E). We then explored the correlation of gene clusters with tyrosine hydroxylase (TH) expression, which is a key pathological mechanism for predicting PD progression and poor prognosis. The results showed that the expression of TH in gene cluster 1 was lower than that in gene cluster 2, indicating that the neuron loss in gene cluster 1 was more significant (Figure 5F). Ultimately, we conducted GO and KEGG enrichment analyses, focusing on the 77 DEGs. Consistent with the GSVA, the results of GO and KEGG showed that these DEGs were enriched in dopaminergic synapse, GABAergic synapse, TCA cycle, etc., which revealed that cuproptosis plays an important role in PD development and progression (Figure 5G,H).

### 3.6. Gene Module Screening and Co-Expression Network Establishment

We implemented the WGCNA to construct a co-expression network based on normal substantia nigra and PD substantia nigra, with the aim of identifying key modules associated with PD. The soft threshold power was set to four, and the scale-free R2 was equal to 0.9 (Figure 6A). Eight modules expressed in different colors were identified by the dynamic tree-cutting algorithm (Figure 6B). Heatmaps were employed to visualize the co-expression relationships between modules and clinical features (normal and PD) (Figure 6C). Among them, the turquoise module had the strongest relationship with PD, including 1384 genes (Cor = −0.43, *p* = 7 × 10^−8^). We found that the turquoise module had a high correlation with PD (Cor = 0.7, *p* < 1 × 10^−200^) (Figure 6D). We also analyzed the key modules associated with CRG clusters using the WGCNA. The soft threshold power was set to four, and the scale-free R2 was equal to 0.9 (Figure 7A). As shown in Figure 7B, 12 co-expression modules were identified through the dynamic tree-cutting algorithm. Co-expression relationships between modules (CRG cluster 1 and CRG cluster 2) and clusters were visualized using heatmaps (Figure 7C). Among the 12 modules, the turquoise module was significantly associated with the CRG clusters (Cor = 0.71, *p* = 2 × 10^−13^). A high correlation was observed between MM and GS in the turquoise module (Cor = 0.87, *p* < 1 × 10^−200^) (Figure 7D).

### 3.7. Identification of Cuproptosis–PD-Related Genes and Functional Analysis

A total of 225 cuproptosis–PD-related genes were identified by the intersection of the CRG cluster module genes and the PD module genes (Figure 8A). The GO and KEGG enrichment analysis was used to further explore the function of the cuproptosis–PD-related genes. The GO enrichment analysis showed that among biological process (BP) categories, the cuproptosis–PD-related genes were enriched in axon development, axonogenesis, modulation of chemical synaptic transmission, and synaptic vesicle cycle (Figure 8B). In cell component (CC) categories, cuproptosis–PD-related genes were enriched in presynapse, synaptic membrane, transport vesicle, and GABA-ergic synapse (Figure 8B). In molecular function (MF) categories, cuproptosis–PD-related genes were enriched in channel activity, tubulin binding, and gated channel activity (Figure 8B). KEGG enrichment analysis results showed that cuproptosis–PD-related genes were enriched in pathways of neurodegeneration, synaptic vesicle cycle, dopaminergic synapse, and cell adhesion molecules (Figure 8C). Therefore, we hypothesized that the cuproptosis–PD-related genes may be involved in the progression of PD through cuproptosis.

### 3.8. Construction of Cuproptosis-Related Machine Learning Models

In order to pinpoint cuproptosis–PD-related genes possessing high diagnostic value, we constructed a LASSO analysis utilizing the cuproptosis–PD-related genes within the PD training cohort. Three genes (KIAA0319, AGTR1, and SLC18A2) were identified as key genes based on the LASSO analysis (Figure 9A,B). A ROC curve was employed to evaluate the performance of the three genes in the training cohort. The highest areas under the ROC curve (AUC) values of the three genes were as follows: KIAA0319 = 0.812, AGTR1 = 0.826, and SLC18A2 = 0.856 (Figure 9C). The boxplot results showed that the three genes were significantly down-regulated in the substantia nigra of PD (Figure 9D–F). Furthermore, we evaluated the correlation between the expression of the three genes and TH expression. We observed a positive correlation between KIAA0319, AGTR1, and SLC18A2 and TH expression (Figure 9G–I).

### 3.9. Establishment of a Nomogram and Evaluation of the Model

A nomogram was constructed to diagnose the PD subtypes using the KIAA0319, AGTR1, and SLC18A2 (Figure 10A). Calibration curves showed a small error between actual and predicted risk for PD clusters (Figure 10B). Subsequently, we validated the three gene prediction models on test cohorts (GSE133101 and GSE26927). The ROC curve showed that the AUC of the three genes predicts well in the test cohort (Figure 10C–E). We built a logistic regression model with an AUC of 0.860 under the ROC curve, indicating that the model had good predictive performance (Figure 10F). Then, we evaluated the expression differences of the top three genes in the total test cohort. The boxplot showed that the expression of KIAA0319, AGTR1, and SLC18A2 in the substantia nigra of PD are all significantly down-regulated, which is consistent with previous results (Figure 10G–I). Correlation analysis showed that the three genes were positively correlated with the expression of TH (Figure 10J–L). The above results proved that the three cuproptosis genes had a certain pathological diagnosis value in PD.

## 4. Discussion

In recent years, due to the rising incidence of PD, gaining a comprehensive understanding of the pathology and molecular mechanisms of PD has become increasingly important for diagnosis and treatment. High-throughput sequencing and bioinformatics analysis have assisted in enhancing our understanding of the molecular mechanisms involved in disease initiation and progression, thereby facilitating the exploration of genetic alterations and the identification of potential diagnostic biomarkers. Current studies have shown that the development of PD is associated with oxidative stress, neuroinflammation, abnormal aggregation of α-synuclein, mitochondrial dysfunction, and cell death [39,40,41]. However, based on current research, the effect of PD treatment is not satisfactory. Therefore, exploring the molecular mechanism is crucial to guide the individualized treatment of PD.

Cuproptosis, as recently proposed, is a novel RCD dependent on copper and mitochondrial respiration [14,42]. The present study has demonstrated that copper binds directly to the lipoylated components of the TCA cycle, leading to the aggregation of lipoylated proteins and the subsequent loss of iron–sulfur cluster proteins, which causes proteotoxic stress and ultimately leads to cell death. The most recent study indicated that copper induces cognitive impairment in mice through the regulation of cuproptosis and CREB signaling pathways [43]. However, the specific mechanisms underlying cuproptosis and its regulatory role in PD remain unclear. In our present study, we first assessed CRGs’ expression in control substantia nigra and PD substantia nigra. Compared with the control population, the CRGs’ expression in the substantia nigra of PD patients is generally abnormal, suggesting that CRGs play an important role in the development of PD. Correlation analysis reveals significant interactions between CRGs in PD, including synergistic and antagonistic effects. Subsequently, we assessed immune cell content between controls and the substantia nigra of PD. The results showed that there were higher levels of naive CD4+ T cells, resting memory CD4+ T cells, resting NK cells, Monocytes, M2 macrophages, and Neutrophils in PD patients, which was consistent with the results of previous studies [44,45,46]. In addition, we used unsupervised cluster analysis to divide 81 PD patients into two clusters to better understand the expression patterns of CRGs. Immunological analysis revealed that cluster 1 exhibited a higher immune score. We also explored the expression of immune checkpoints between the two clusters, and the results showed that the expression of immune checkpoint-related genes in cluster 1 was generally increased, which supported the above results [47]. GSVA and GSEA analysis indicated that CRG cluster 1 was enriched in the Notch pathway, ECM receptor interaction, JAK/STAT pathway, and cytokine receptor interaction, while CRG cluster 2 was significantly enriched in cell cycle, ubiquitin-mediated proteolysis, pyruvate metabolism, and TCA cycle. It has been reported that the Notch signaling pathway and the JAK/STAT signaling pathway are involved in neurodegeneration and neuroinflammation in PD, respectively [48,49]. Taken together, we have reason to believe that cluster 1 may have more neuroimmune involvement in the progression of PD.

To further explore the biology of CRG clusters, we performed differential analysis and identified DEGs between CRG clusters. Based on the DEGs, we performed unsupervised clustering to classify PD into two gene clusters. The heatmap results showed that the gene clusters were generally consistent with the CRG cluster classification. Tyrosine hydroxylase (TH) is the initial and rate-limiting enzyme in the biosynthesis of dopamine, and its dysregulation is a key pathological mechanism in predicting PD progression and poor prognosis [50,51]. The boxplot analysis showed that TH expression was lower in gene cluster 1 than in gene cluster 2, predicting more severe neuronal damage in gene cluster 1. The enrichment analysis showed that DEGs were enriched in dopaminergic synapses, GABAergic synapses, and the TCA cycle, indicating that CRGs play an important role in the development of PD.

In recent years, WGCNA algorithms and machine learning models have been increasingly used in the screening of PD-related genes and the prediction of disease prevalence [37]. We used the WGCNA algorithm to identify key modules related to PD and CRG and finally obtained 225 intersection genes. Enrichment analysis of these intersection genes showed that cuproptosis–PD-related DEGs were enriched in certain pathways, such as neurodegeneration, synaptic vesicle cycle, dopaminergic synapse, and cell adhesion molecules. The results show that the cuproptosis–PD-related DEGs are mainly expressed in the brain and play an important role in maintaining the transmission of nerve signals in the brain, indicating involvement in the development of neurodegeneration. In this study, we conducted a LASSO analysis based on the expression profiles of cuproptosis–PD-related DEGs. Subsequently, we selected the top three genes (KIAA0319, AGTR1, and SLC18A2) for further investigation. KIAA0319 encodes a transmembrane protein that regulates neuronal migration and cell adhesion in the central nervous system, but the exact mechanism in PD remains unclear [52]. AGTR1, also known as angiotensin II receptor type 1, has been extensively studied in tumors and neurodegenerative diseases. Tushar et al. found that AGTR1 is spatially restricted to the ventral side of SNpc and highly susceptible to loss in PD [53]. SLC18A2 is an integral membrane protein that mediates the transport of monoamine neurotransmitters from the cytoplasm to synaptic vesicles. Dysfunctions of SLC18A2 have been suggested to contribute to the pathogenesis of PD [54]. Then, we constructed a nomogram model to diagnose the PD subtype using KIAA0319, AGTR1, and SLC18A2. Two validation cohorts (GSE133101 and GSE26927) were used to evaluate the efficacy of the three genes, and the ROC results showed that the model had good predictive performance and certain clinical application values. Additionally, we used the test cohorts to detect the expression of the three genes between the control and PD. The results are consistent with the training set. Correlation analysis showed that the three genes have a positive correlation with TH expression, suggesting that these genes are involved in maintaining neuronal function in PD.

Although many novel explorations have been carried out in this study, there are still some limitations. Firstly, to improve the accuracy of the model, more detailed clinical characteristics should be included in the analysis. Furthermore, more PD samples need to be included to ensure the accuracy of copper-poisoning-associated clusters, and the potential correlation between CRGs and immune infiltration needs further study. Finally, this study needs further analyses to validate the expression pattern of CRGs in PD patients.

## 5. Conclusions

In summary, our research has revealed the expression patterns of cuproptosis-related genes in PD through transcriptome integration analysis. By dividing CRGs into two clusters, we found that immune cell infiltration and signaling pathways were different in the two CRG clusters. Through machine learning, we identified KIAA0319, AGTR1, and SLC18A2 as the core genes involved in the pathogenesis of PD. Furthermore, we have established a prediction model that can accurately assess the risk of PD. Our study systematically evaluated the relationship between cuproptosis and PD, which will help to improve the understanding of cuproptosis in PD.

## Figures and Tables

**Figure 1 brainsci-13-01015-f001:**
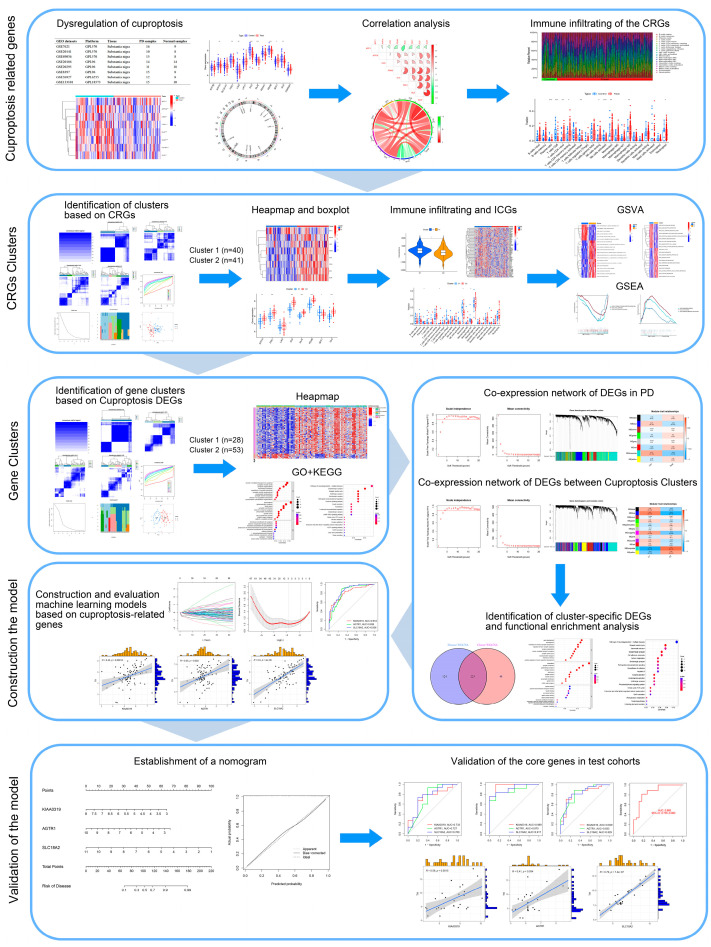
Flowchart of this study. * *p* < 0.05; ** *p* < 0.01; *** *p* < 0.001.

**Figure 2 brainsci-13-01015-f002:**
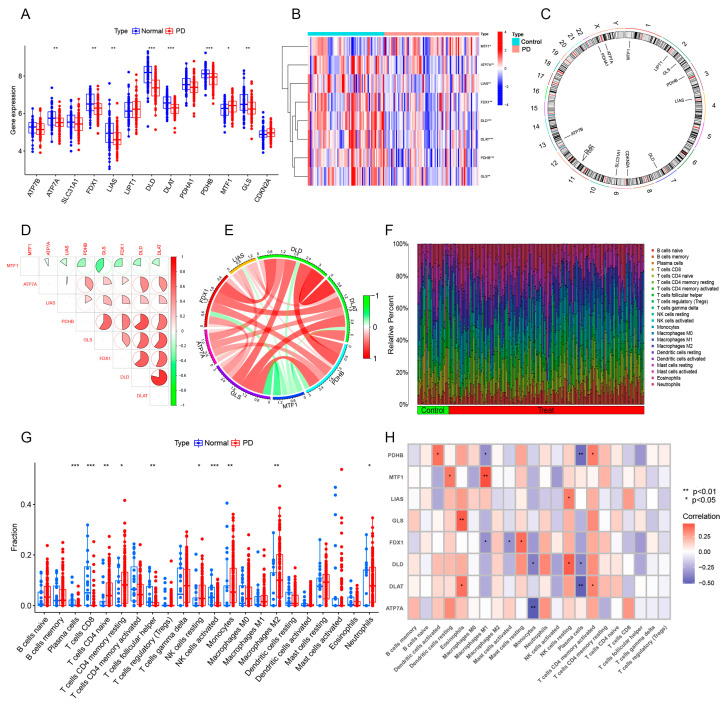
Expression distributions and immune infiltrating landscape of CRGs in PD. (**A**) The expression of 13 CRGs between PD substantia nigra and normal substantia nigra. (**B**) Heatmap showing the expression pattern of differentially expressed CRGs. (**C**) Circus plot showing the location of the 13 CRGs on chromosome. (**D**,**E**) The correlation of 12 differentially expressed CRGs. Blue and red colors represent positive and negative correlations, respectively. (**F**) The abundance of immune infiltrating between PD substantia nigra and normal substantia nigra. (**G**) Boxplot showing the difference in immune infiltration between PD substantia nigra and normal substantia nigra. (**H**) The correlation of 8 differentially expressed CRGs with immune infiltration. * *p* < 0.05; ** *p* < 0.01; *** *p* < 0.001.

**Figure 3 brainsci-13-01015-f003:**
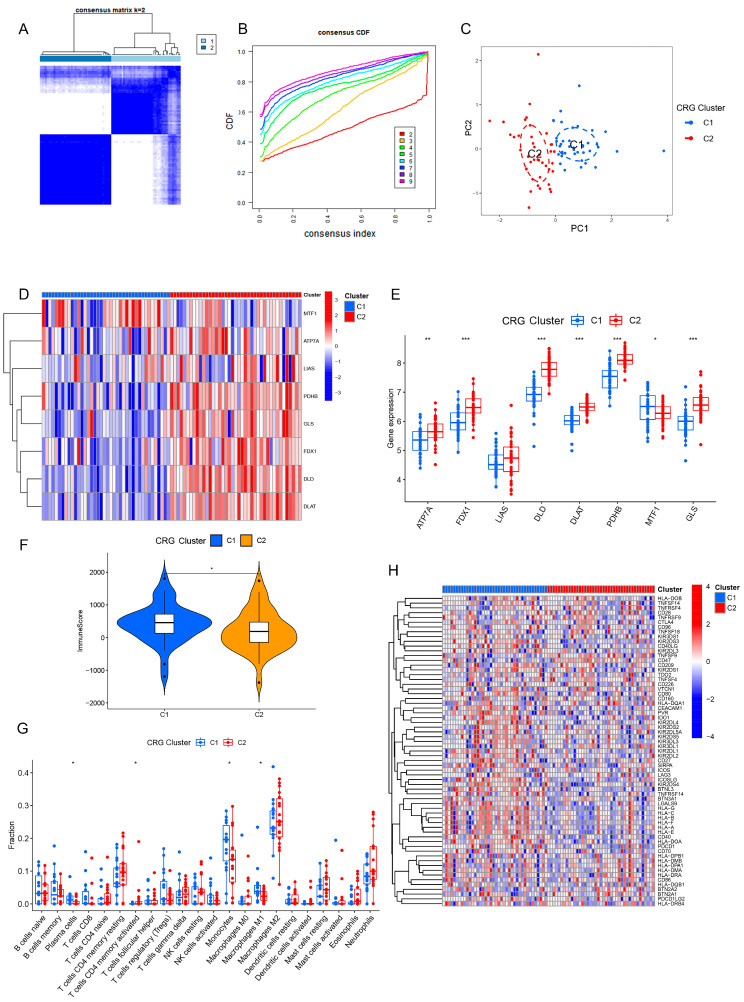
Identification of CRG clusters in PD. (**A**,**B**) PD patients were divided into two groups according to the consensus score matrix (k = 2) and CDF plots. (**C**) PCA plot of CRG clusters. (**D**) Heatmap showing expression patterns of 8 CRGs in different CRG clusters. (**E**) The expression of 8 CRGs in different CRG clusters. (**F**) The differences in immune scores between the 2 CRG clusters. (**G**) Boxplot showing the difference in immune infiltration between the 2 CRG clusters. (**H**) Heatmap showing the differential expression of immune checkpoint genes between the 2 CRG clusters. * *p* < 0.05; ** *p* < 0.01; *** *p* < 0.001.

**Figure 4 brainsci-13-01015-f004:**
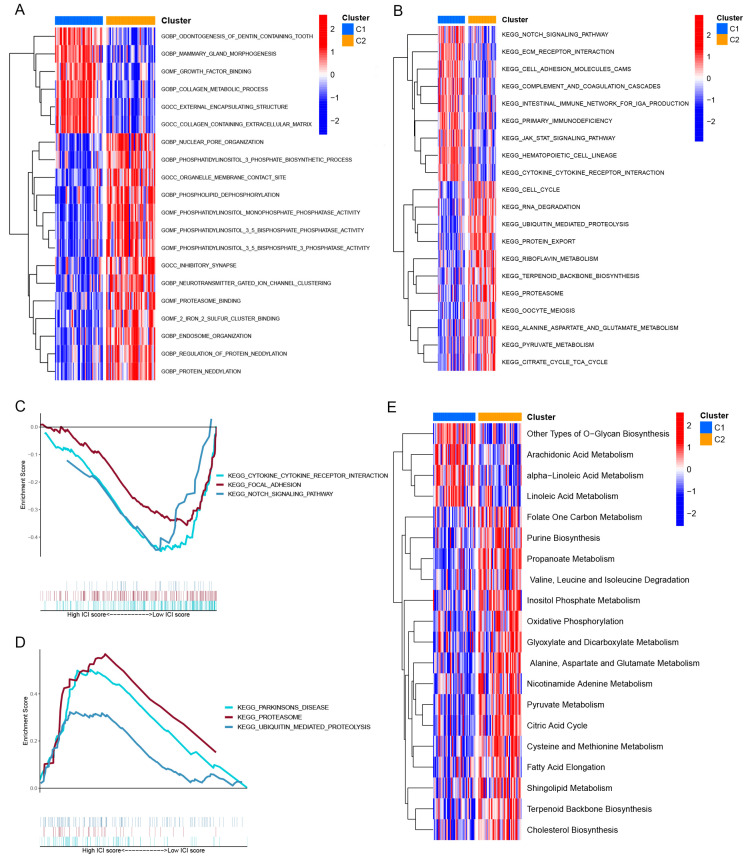
Functional enrichment analysis between CRG clusters. (**A**) GSVA analysis of GO terms between the 2 CRG clusters. (**B**) GSVA analysis of KEGG terms between the 2 CRG clusters. (**C**) GSEA analysis of CRG cluster 1. (**D**) GSEA analysis of CRG cluster 2. GSVA, gene set variation analysis; GO, Gene Ontology; KEGG, Kyoto Encyclopedia of Genes and Genomes; GSEA, gene set enrichment analysis. (**E**) Heatmap showing the enrichment of metabolism-related signatures between the 2 clusters.

**Figure 5 brainsci-13-01015-f005:**
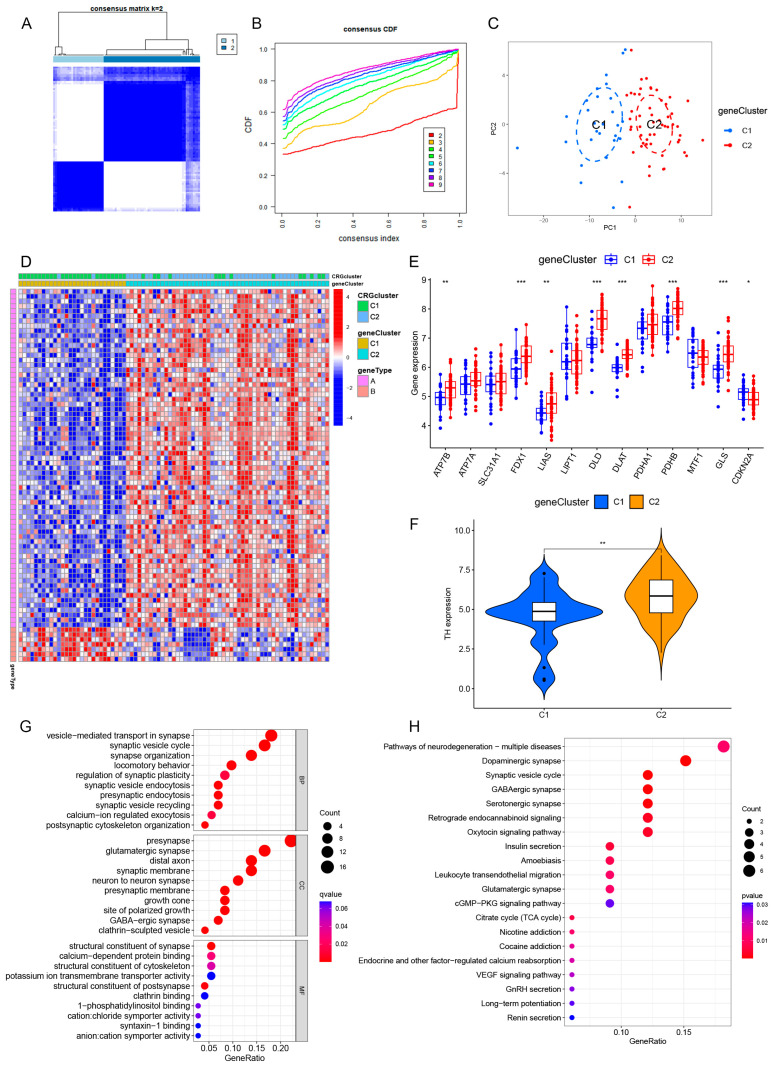
Identification of cuproptosis gene subtypes in PD. (**A**,**B**) Differential expression genes (DEGs) were identified between CRG clusters, and DEGs were divided into two groups according to the consensus score matrix (k = 2) and CDF plot. (**C**) PCA plot of the gene clusters. (**D**) Heatmap showing the differential expression of DEGs and cuproptosis characteristics between gene clusters 1 and 2. (**E**) The expression of CRGs between gene clusters 1 and 2. (**F**) The correlation of gene clusters with TH expression. (**G**) GO analysis between gene clusters 1 and 2. (**H**) KEGG analysis between gene clusters 1 and 2. TH, tyrosine hydroxylase; GO, Gene Ontology; KEGG, Kyoto Encyclopedia of Genes and Genomes. * *p* < 0.05; ** *p* < 0.01; *** *p* < 0.001.

**Figure 6 brainsci-13-01015-f006:**
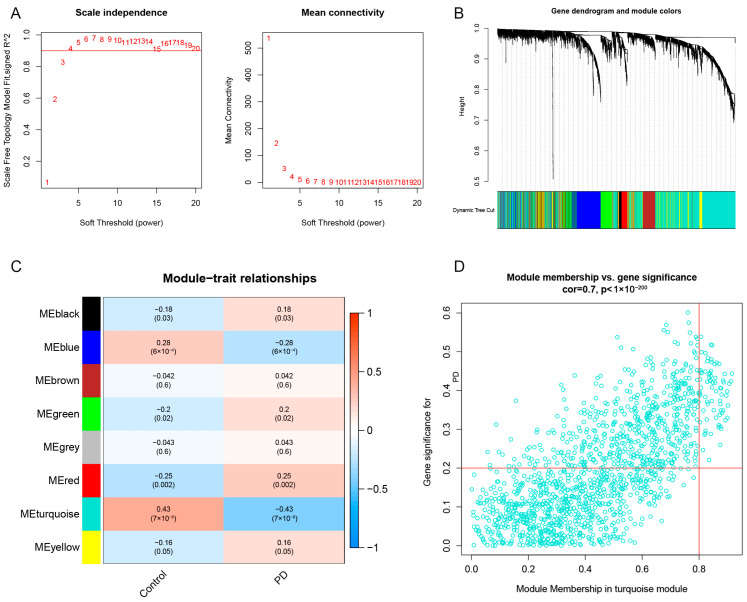
Identification of key modules between PD and control. (**A**) The selection of soft threshold power. (**B**) Co-expression module clustering tree dendrogram. Different colors represent different co-expression modules. (**C**) Correlation heatmap analysis between different modules and PD. (**D**) Scatter plot between turquoise modules and PD.

**Figure 7 brainsci-13-01015-f007:**
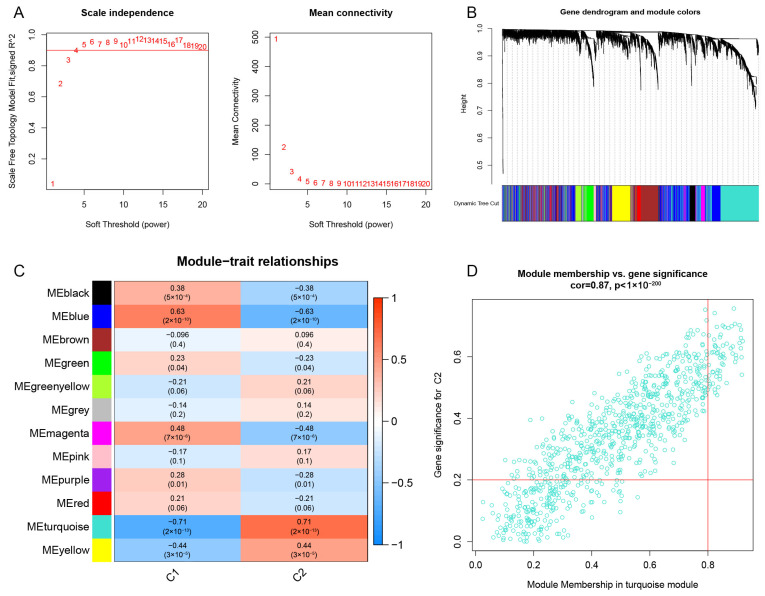
Identification of key modules between the two cuproptosis clusters. (**A**) The selection of soft threshold power. (**B**) Co-expression module clustering tree dendrogram. Different colors represent different co-expression modules. (**C**) Correlation heatmap analysis between different modules and CRG clusters. (**D**) Scatter plot between turquoise modules and cuproptosis cluster 2.

**Figure 8 brainsci-13-01015-f008:**
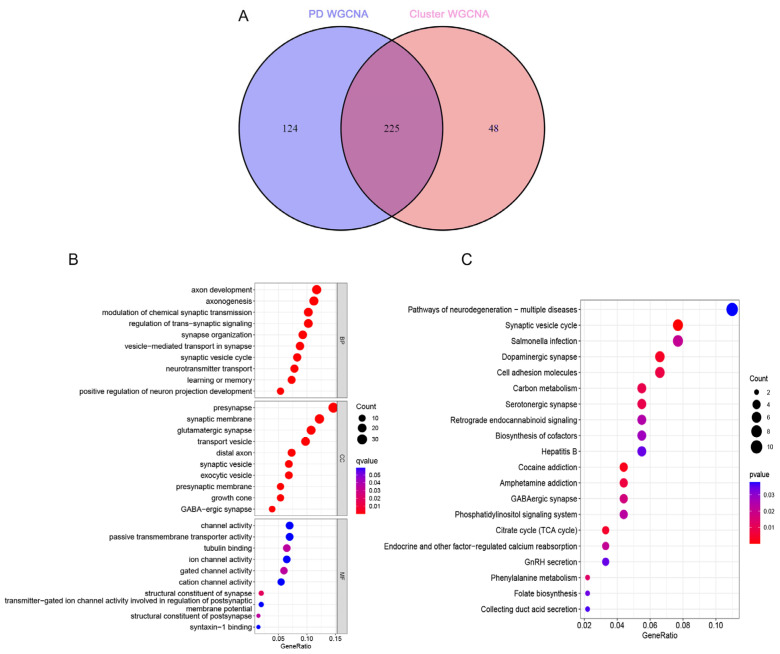
Identification of cuproptosis–PD-related genes and functional analysis. (**A**) The intersections between PD-related module genes and cuproptosis cluster module genes. (**B**) The GO analysis of the cuproptosis–PD-related genes. (**C**) The KEGG analysis of the cuproptosis–PD-related genes. GO, Gene Ontology; KEGG, Kyoto Encyclopedia of Genes and Genomes.

**Figure 9 brainsci-13-01015-f009:**
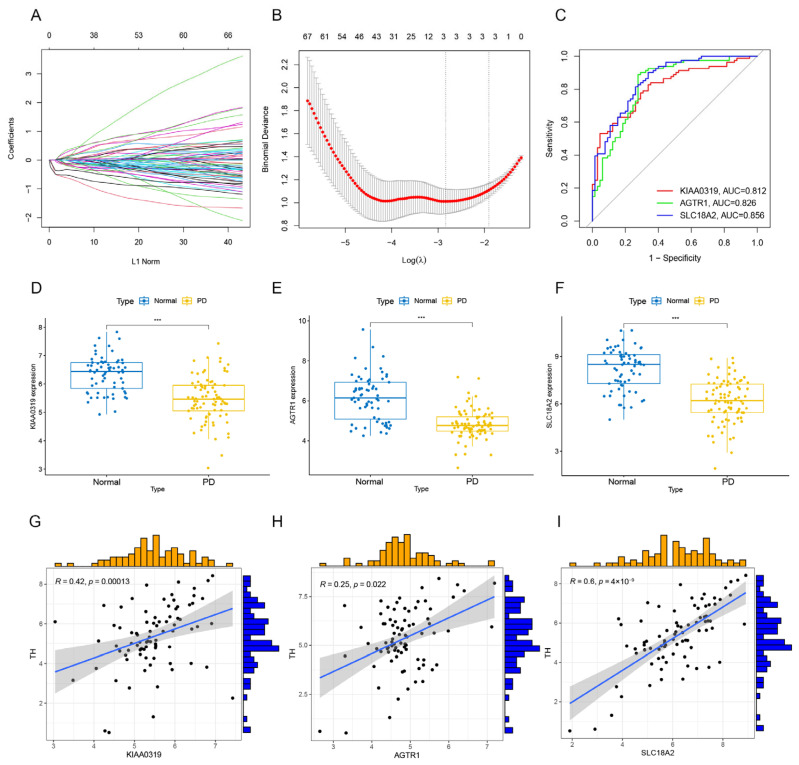
Construction and evaluation of machine learning models based on LASSO analysis. (**A**,**B**) We performed LASSO regression on the DEGs and plotted the cross-validation curve for selecting the tuning parameter (λ). (**C**) The ROC curve of the core genes. (**D**–**F**) The boxplots showing the 3 genes’ (SLC18A2, KIAA0319, and AGTR1) expression in PD substantia nigra and normal substantia nigra. (**G**–**I**) The correlation analysis between the TH expression and the core genes. *** *p* < 0.001.

**Figure 10 brainsci-13-01015-f010:**
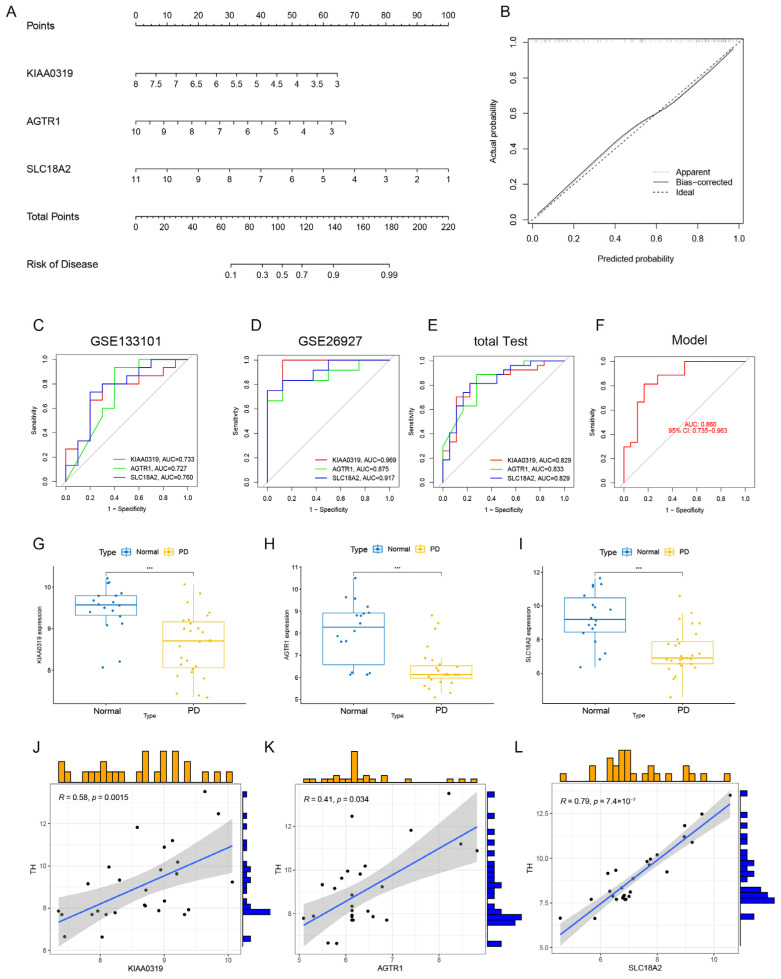
Establishment of a nomogram and evaluation in the test cohort. (**A**) Construction of a nomogram based on the core genes for predicting the risk of PD. (**B**) Construction of calibration curve. (**C**) The ROC curve in GSE133101. (**D**) The ROC curve in GSE26927. (**E**) The ROC curve in the total test cohort. (**F**) The ROC curve of the model in the total test cohort. (**G**–**I**) Boxplots showing the core genes’ expression in PD substantia nigra and normal substantia nigra in the total test cohort. (**J**–**L**) Correlation analysis between the TH expression and the core genes in the total test cohort. *** *p* < 0.001.

**Table 1 brainsci-13-01015-t001:** The information of the GEO datasets.

GEO Datasets	Platform	Tissue	PD Samples	Normal Samples
GSE7621	GPL570	Substantia nigra	16	9
GSE20141	GPL570	Substantia nigra	10	8
GSE49036	GPL570	Substantia nigra	15	8
GSE20186	GPL96	Substantia nigra	14	14
GSE20295	GPL96	Substantia nigra	11	18
GSE8397	GPL96	Substantia nigra	15	8
GSE26927	GPL6255	Substantia nigra	12	8
GSE133101	GPL18573	Substantia nigra	15	10

## Data Availability

The datasets (GSE7621, GSE20141, GSE49036, GSE20186, GSE20295, GSE8397, GSE26927, and GSE133101) for this study can be found in the GEO (http://www.ncbi.nlm.nih.gov/geo/ (accessed on 13 November 2022)) database. The code applied in this study is available from the corresponding author upon reasonable request.

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
