# Peer review of "Identification of Cuproptosis Clusters and Integrative Analyses in Parkinson’s Disease"

_brainsci, 2023, doi:10.3390/brainsci13071015_

Round 1

Reviewer 1 Report

- figures are not clear at all, try to make them more readable.

- conclusion must be more detailed.

- use recent references in discussion. 

- add more numbers to introduction, about parkinson disease. 

Author Response

Point 1:  figures are not clear at all, try to make them more readable.

Response 1: Thank you for your valuable suggestions. In response to the reviewer's feedback, we have made adjustments to the font size in our figures to enhance readability.

Point 2: conclusion must be more detailed.

Response 2: Thank you for your valuable suggestions. We have taken the reviewer's advice and added more details to the Conclusions section.

Point 3: use recent references in discussion.

Response 3: Thanks for your suggestions. Based on the reviewer's suggestion, we have made the following adjustments in the Discussion section: (1) removed earlier references in the Discussion section; (2) relocated lines 388-404 to the Introduction section (3) added more descriptive content.

Point 4: add more numbers to introduction, about parkinson disease.

Response 4: Thanks for your suggestions. Following the reviewer's suggestion, we have added more Parkinson's disease content in the Introduction section.

I would like to sincerely thank you for all your help and support, and I am eagerly looking forward to hearing from you soon.

Reviewer 2 Report

This study is intriguing since it provides a fresh knowledge of PD-related molecular pathways as well as new insights into PD treatment. The following minor issues should be addressed.

1.     The paragraph from line 388-404 needs to be transferred to the introduction.

2.     Line 162-164 should be deleted.

3.     The resolutions of the figures need to be improved.

4.      The detailed clinical characteristics of Parkinson's disease patients are missing.

5.     The sample size included in the analysis is small to draw up the reported conclusion of the study.

6.     The language needs to be revised.

The language needs to be revised.

Author Response

Thank you very much for your comments on the manuscript and we revised it seriously. Follows are the point-to-point response.

Point 1: The paragraph from line 388-404 needs to be transferred to the introduction.

Response 1: Thank you for your valuable suggestions. Following the reviewer's suggestion, we moved the paragraph on lines 388-404 to the Introduction section.

Point 2: Line 162-164 should be deleted.

Response 2: Thanks for your suggestions. I have deleted line162-164.

Point 3: The resolutions of the figures need to be improved.

Response 3: Thank you for your valuable suggestions. In response to the reviewer's feedback, we have made adjustments to the font size in our figures to enhance readability.

Point 4: The detailed clinical characteristics of Parkinson's disease patients are missing.

Response 4: Thank you very much for the reviewer's professional suggestions. The samples utilized in this study were obtained from the GEO database. In this study, only a few datasets included gender and age information, while the majority of the datasets lacked clinical characteristics. Due to the lack of clinical information, we were unable to perform an integrated analysis of transcriptomic and clinical features.

Point 5: The sample size included in the analysis is small to draw up the reported conclusion of the study.

Response 5: Thank you for your valuable suggestions. Based on the reviewer's feedback, we have revised the conclusion section. Please kindly review it.

Point 6: The language needs to be revised.

Response 6: According to the reviewer's recommendation, we have requested our colleagues who possess fluency in English to conduct a thorough examination of this manuscript.

I would like to sincerely thank you for all your help and support, and I am eagerly looking forward to hearing from you soon.

Reviewer 3 Report

The manuscript ˝Identification of cuproptosis clusters and integrative analyses in Parkinson’s disease˝ by Zhang et al. is the bioinformatics analysis of relationship between cuproptosis-related genes and immune profile in Parkinson's desease. Numerous interrelation were tested and relevant results were obtained. The manuscript is well organized. Methods and statystical analyses are appropriate. References are also appropriate and recent.

I suggest to the authors to make only a few minor corrections:

-        - p.5, line 185 – change NK cells resting  to resting NK cells

-        - p.5, line 185/186 - change Macrophages M2  to M2 macrophages

-        - p.5, line 186 - change T cells follicular helper to follicular helper T cells or toT follicular helper cells

-        - the previous three suggestions also refer to the discussion

-        - p.10, line 274 – change Th to TH which is used in rest of the manuscript

-       -  p.14, line 359 – change GSE26727 to GSE26927

-        - p.14, line 362 – change Figures 10F to Figure 10F

-        - p.14, line 365 – change Figures 10G-J to Figures 10G-I

-      -  a general remark refers to the size of the letters (too small) on some parts of the figures, especially the titles of the axes on diagrams

Author Response

Thank you very much for your comments on the manuscript and we revised it seriously. Follows are the point-to-point response.

Point 1: p.5, line 185 – change NK cells resting  to resting NK cells

Response 1: Thanks for your suggestions. Based on the reviewer's suggestion, I have revised NK cells resting to resting NK cells.

Point 2: p.5, line 185/186 - change Macrophages M2  to M2 macrophages

Response 2: Thanks for your suggestions. Based on the reviewer's suggestion, I have revised Macrophages M2 to M2 macrophages.

Point 3: p.5, line 186 - change T cells follicular helper to follicular helper T cells or to T follicular helper cells

Response 3: Thanks for your suggestions. Based on the reviewer's suggestion, I have revised T cells follicular helper to follicular helper T cells.

Point 4: the previous three suggestions also refer to the discussion

Response 4: Based on the reviewer's suggestion, I have revised the previous three suggestions in the discussion section.

Point 5: p.10, line 274 – change Th to TH which is used in rest of the manuscript

Response 5: Thanks for your suggestions. I have revised Th to TH.

Point 6: p.14, line 359 – change GSE26727 to GSE26927

Response 6: Thanks for your suggestions. I have revised GSE26727 to GSE26927.

Point 7: p.14, line 362 – change Figures 10F to Figure 10F

Response 7: Thanks for your suggestions. I have revised Figures 10F to Figure 10F.

Point 8: p.14, line 365 – change Figures 10G-J to Figures 10G-I

Response 8: Thanks for your suggestions. I have revised Figures 10G-J to Figures 10G-I.

Point 9: a general remark refers to the size of the letters (too small) on some parts of the figures, especially the titles of the axes on diagrams

Response 9: Thank you for your valuable suggestions. We have made adjustments to the font size in our figures in response to the reviewer's feedback, with the aim of enhancing readability.

I would like to sincerely thank you for all your help and support, and I am eagerly looking forward to hearing from you soon.